

# Near real time processing of ceilometer network data: characterizing an extraordinary dust outbreak over the Iberian Peninsula

Alberto Cazorla[1,2], Juan Andrés Casquero-Vera[1,2], Roberto Román[1,2], Juan Luis Guerrero-Rascado[1,2], Carlos Toledano[3], Victoria E. Cachorro[3], José Antonio G. Orza[4], María Luisa Cancillo[5,6], Antonio Serrano[5,6], Gloria Titos[7], Marco Pandolfi[7], Andres Alastuey[7], Natalie Hanrieder[8], Lucas Alados-Arboledas[1,2]

[1]Andalusian Institute for Earth System Research, IISTA-CEAMA, University of Granada, Junta de Andalucía, Granada, Spain
[2]Department of Applied Physics, University of Granada, Granada, Spain
[3]Grupo de Óptica Atmosférica (GOA), Universidad de Valladolid, Valladolid, Spain
[4]SCOLAb, Física Aplicada, Universidad Miguel Hernández, Elche, Spain
[5]Departament of Physics, University of Extremadura, Badajoz, Spain
[6]Institute of Water Research, Climate Change and Sustainability, IACYS. University of Extremadura, Spain
[7]Institute of Environmental Assessment and Water Research (IDAEA-CSIC), Barcelona, Spain
[8]German Aerospace Center (DLR), Institute of Solar Research, Plataforma Solar de Almería, Almería, Spain

*Correspondence to*: Alberto Cazorla (cazorla@ugr.es)

**Abstract.** The interest on the use of ceilometers for optical aerosol characterization has increased in the last few years. They operate continuously almost unattended and are also much less expensive than lidars, hence they can be distributed in dense networks over large areas. However, due to the low signal-to-noise-ratio it is not always possible to obtain particle backscatter coefficient profiles, and the vast amount of data generated requires an automated and unsupervised method that ensures the quality of the profiles inversions.

In this work a method that uses aerosol optical depth (AOD) measurements from the AERONET network is applied for the calibration and automated quality assurance of inversion of ceilometer profiles. The method is compared with independent inversions obtained by co-located multiwavelength lidar measurements and a difference up to 15% in backscatter is found between both instruments. This method is continuously and automatically applied to the Iberian Ceilometer Network (ICENET) and a case example during an unusually intense dust outbreak affecting the Iberian Peninsula on 20 February 2016 and lasted until 24 February 2016 is shown. Results reveal that it is possible to obtain a quantitative optical aerosol characterization (particle backscatter coefficient) with ceilometers over large areas and this information has a great potential for alert systems and model assimilation and evaluation.

## 1 Introduction

Atmospheric aerosol is one of the main responsible factors of climate radiative forcing through multiple processes including aerosol-radiation and aerosol-cloud interactions (IPCC, 2014). The aerosol direct effects depend on their optical properties,





and also on their spatial and vertical distribution in the atmosphere. In spite of the recent advances on instrumentation that has improved the ability of characterizing key aerosol properties and their spatial resolution, the associated uncertainties are still considered to be one of the majors in climate forcing (Boucher et al., 2013).

In this sense, the implementation of observational networks is crucial for spatial characterization of aerosol properties.

Ground-based networks represent key tools in the study of aerosol radiative forcing. In-situ observational networks provide surface measurements distributed over large areas, e.g. the Global Atmospheric Watch, GAW (GAW, 2011) and ACTRIS (www.actris.eu) for the European Union. On the other hand, one of the recognized instruments for the retrieval of column-integrated aerosol properties is the robotic sun and sky photometer that is used in the global Aerosol Robotic NETwork (AERONET; Holben et al. 1998; Dubovik et al. 2006). Lidar systems are well known active remote sensing instruments for

the vertical resolved characterization of aerosol optical and microphysical properties (Winker et al., 2003). GAW Atmospheric Lidar Observation Network (GALION) has emerged as an initiative of the GAW aerosol program (GAW, 2008). Its main objective is to provide the vertical component of the aerosol distribution through advanced laser remote sensing in a network of ground-based stations. Among other networks, GALION includes the European Aerosol Research Lidar Network (EARLINET) that provides vertical aerosol profile observations over Europe based on 27 instruments in 16

countries (Pappalardo et al. 2014), the Micro Pulse Lidar Network, MPLNET (Welton et al. 2001), and the Latin American Lidar Network, LALINET (Guerrero-Rascado et al. 2016).

In order to obtain a larger spatial coverage than ground-based networks, in the last few years some space missions have been promoted focusing on aerosol measurements from satellites, e.g the Lidar in Space Technology Experiment, LITE (McCormick, 1997) and the Cloud-Aerosol Lidar and Infrared Pathfinder Satellite Observation, CALIPSO (Winker et al.

2003). The main disadvantage of measurements from space-borne platforms is the low temporal resolution, since the measurements are limited to the satellite passes over a region. Furthermore, the quality of the products offered from these sensors is still inadequate (e.g. Li et al. 2009) and ground-based instrumentation are needed for validation.

The usefulness of vertical resolved aerosol characterization has been proven by monitoring dust outbreaks (e.g. Guerrero-Rascado et al, 2008; 2009; Cordoba-Jabonero et al. 2011; Bravo-Aranda et al. 2015; Preissler et al. 2011; 2013; Granados-

25 Muñoz et al. 2016; Valenzuela et al. 2017), biomass burning plumes (e.g. Alados-Arboledas et al. 2011; Ortiz-Amezcua et al. 2016), or the volcanic ash plume from the Eyjafjallajökull eruptionon April 2010 (Navas-Guzmán et al., 2013; Pappalardo et al. 2013; Sicard et al.,2012). Precisely, this singular event caused aviation problems and drew the attention to the use of ceilometers for vertical resolved aerosol characterization (Adam et al. 2016).

The complexity of lidar systems requires staff to be trained in their operation and the analysis procedures are not fully

automated in many stations. In this sense, continuous operation of lidar systems is not feasible for most stations. In addition, economic and operational costs make complicate the implementation of dense lidar networks. On the other hand, ceilometers are one-wavelength (near infrared) lidars with simple technical specifications (eye-safe low pulse energy and high pulse repetition frequencies) allowing for unattended and continuous operation. Originally designed for cloud base determination, their performance has been improved in the last few years. It has been shown their capabilities for determining planetary




boundary layer (e.g., Wiegner et al., 2006; Münkel et al., 2007; Haeffelin et al., 2011; Pandolfi et al., 2013), detection and forecast of fogs (Haeffelin et al., 2016), and recent efforts have been conducted to quantify the aerosol optical information derived from ceilometers (Frey et al., 2010; Heese et al., 2010, Wiegner et al., 2014).

The main advantage of the use of ceilometers for aerosol characterization is, on the one hand, the automatic and much simpler operation compared to lidars that allows a continuous operation providing range corrected signal profiles 24 hours a day and, on the other hand, the possibility of installing them distributed over large areas. Meteorological services such as those in Germany, France, the Netherlands or United Kingdom are deploying ceilometers networks to cover their national territories with the objective of reaching a spatial density of nearly one device every 100km (e.g. Haij and Klein-Baltink 2007; Flentje et al. 2010). Due to a dense number of instruments and continuous measurements, operative networks need an automated processing and a protocol that ensures the quality of the data.

In this sense, two programs in Europe are dealing with the use of automated lidars and ceilometers for aerosol and cloud properties characterization. The COST Action ES1303 TOPROF (TOwards operational ground based PROFiling with ceilometers, doppler lidars and microwave radiometers for improving weather forecasts) aims in one of its working groups at better characterizing the parameters that can be derived from ceilometer measurements and related uncertainties. At the same time, E-PROFILE, a program of EUMETNET (EUropean METeorological services NETwork), focuses on the harmonization of ceilometer measurements and data provision across Europe.

In this study we present the implementation of procedures to manage a regional ceilometer network for aerosol characterization over the Iberian Peninsula, the Iberian Ceilometer Network (ICENET). An automatic calibration procedure is applied to the ceilometers and this calibration is used to validate the elastic inversion automatically applied to the profiles in order to obtain particle backscatter coefficient profiles. This method uses additional aerosol optical depth (AOD) information during the calibration for the quality assurance of the data and all processes are performed in near real-time. The capabilities of this distributed network are explored by characterizing an unusually intense dust outbreak affecting the Iberian Peninsula on 20 February 2016 until 24 February 2016. Additionally, a multi-wavelength (MW) Raman lidar is used to validate the retrievals from ceilometers and the quality assurance procedure.

The objective of this study is to obtain reliable vertical resolved aerosol optical properties with ceilometers in near real-time, with special emphasis on strong events, such as mineral dust outbreaks, volcanic plumes, severe biomass burning episodes or contamination episodes. Thus, the aerosol information can be potentially used as an alert system for aviation or weather services, or feed models for assimilation and validation.

## 2 Instrumentation: the Iberian Ceilometer Network

An initiative of the Atmospheric Physics Group of the University of Granada has been the coordination of a network of ceilometers combined with sun-sky photometers (Iberian Ceilometer Network, ICENET) for the characterization of atmospheric aerosol with the objective of obtaining reliable vertically resolved aerosol optical properties in near real-time.



On a first stage, the goal is obtaining the total attenuated backscatter for all ceilometers in the network, i.e. to obtain comparable output from ceilometers and, on a second stage, applying an inversion algorithm to the ceilometer profiles in order to obtain the particle backscatter coefficient. All sites of this new network have a co-located AERONET CE318 sun-sky photometer (Cimel Electronique) that is used to constraint the ceilometers inversion retrievals. In addition, the high

performance lidar system located at the EARLINET Granada station is used as an independent validation of the inversions. This nested approach combining high performance systems like those operated in EARLINET and the distributed ceilometer plus sun-sky photometer is an example of synergy among active and passive remote sensing observations in ACTRIS research infrastructure (www.actris.eu).

Figure 1 shows a map of the ceilometer distribution over the Iberian Peninsula, and Table 1 presents the characteristics of
each site.

All sites operate a Jenoptik (now Lufft) CHM15k-Nimbus ceilometer and have a co-located AERONET sun-sky photometer, except Montsec station (MSA) that has the photometer 770 m above the ceilometer and at a horizontal distance of 2 km approximately. The ceilometer at Murcia (UMH) was not operative during the outbreak studied in this work.

The CHM15k is a ceilometer that operates with a pulsed Nd:YAG laser emitting at 1064 nm. The energy per pulse is 8.4 µJ
with a repetition frequency in the range of 5 – 7 kHz. The laser beam divergence is less than 0.3 mrad and the laser backscattered signal is collected on a telescope with a field of view of 0.45 mrad. The signal is detected by an avalanche photo diode in photon counting mode. Complete overlap of the telescope and the laser beam is found about 1500 m above the instrument (Heese et al., 2010). According to the overlap function provided by the manufacturer, the overlap is 90% complete at 555 m agl. The vertical resolution used is 15 m and the maximum height recorded is 15360 m agl. Ceilometers at
Granada (UGR), Tabernas (PSA) and Valladolid (UVA) operate at a temporal resolution of 15 s while ceilometers at Montsec (MSA) and Badajoz (UEX) operate at a temporal resolution of 1 min.

The process of calibration for ceilometers described on the next section is assisted with AOD data from co-located AERONET stations. All sun-sky photometers near the ceilometers belong to the Iberian network for aerosol measurements (RIMA), a regional network federated to AERONET. This means that all instruments are routinely calibrated following the
same protocol and the data are quality controlled. The sun-photometer provides solar extinction measurements at 340, 380, 440, 675, 870, 936 and 1020 nm, allowing for computing the AOD at these wavelengths (except 936 nm). The AOD uncertainty ranges from ±0.01 in the infrared-visible to ±0.02 in the ultraviolet channels (Holben et al., 1998). For comparison with the ceilometers the AOD is extrapolated to 1064 nm by the Angström law (Angström, 1964) using the AOD measurements at 870 and 1020 nm. Level 1.5 AERONET data, which are automatically cloud-screened and delivered
in near real time, are used in this analysis.

At UGR station a multi-wavelength Raman lidar system (MULHACEN) is used for validation of the ceilometer inversions. The upgraded LR331-D400 (Raymetrics Inc.) operated at IISTA-CEAMA (Andalusian Institute for Earth System Research) is part of EARLINET since April 2005. This lidar system is a ground-based, six wavelength system with a pulsed Nd:YAG laser. The emitted wavelengths are 355, 532 and 1064 nm with output energies per pulse of 60, 65 and 110 mJ, respectively.





It has elastic backscatter channels at 355, 532 and 1064 nm and Raman-shifted channels at 387 (from $N_2$), 408 (from $H_2O$) and 607 nm (from $N_2$). Full overlap is reached around 1220 m agl although the overlap is complete at 90% between 520 and 820 m agl (Navas-Guzmán et al., 2011). Appropriate overlap corrections are derived following the procedure of Wandinger et al. (2002).

## 3 Methodology

Elastic lidars and ceilometers principle of measurement is the same and retrieving optical properties in both systems follow the lidar equation (the dependency with the wavelength has been omitted for simplicity since it is always the same in ceilometers):

$$P(z) = C_L^* \cdot \frac{O(z)}{z^2} \beta(z) \cdot T^2(z). \tag{1}$$

where $P(z)$ is the backscattered power received in the telescope from a distance z, $C_L^*$ is a constant that depends on the geometry and characteristics of the instrument and universal constants, and the term $z^2$ accounts for the acceptance solid angle of the receiver optics with the distance to the laser. The backscattered signal collected by the telescope depends on the overlap between the laser beam and the telescope field of view, and the degree of overlap is quantified by $O(z)$, ranging from 0, if there is not overlap, to 1, if overlap is complete. $\beta(z)$ is the atmospheric backscatter coefficient and $T(z)$ estimates the atmospheric transmittance of the laser signal (squared due to travel back and forth). Also, both properties can be split into contributions of particles and molecules ($\beta(z) = \beta_m(z) + \beta_p(z)$; $T(z) = T_m(z) \cdot T_p(z)$) (Fernald, 1984).

On Eq. (1) the only properties depending on the medium are $\beta(z)$ and $T(z)$. Thus, the atmospheric attenuated backscatter is defined as:

$$\beta_{att}(z) = \beta(z) \cdot T^2(z) \tag{2}$$

The ceilometers used in this study provides as output the range corrected signal ($RCS(z) = P(z) \cdot z^2$), and also, the overlap function of the instrument is factory determined. Therefore, the only parameter that needs to be addressed is the constant $C_L^*$. Wiegner et al. (2014) describes a method to find the $C_L^*$ constant in ceilometers, commonly referred as ceilometer calibration. This method compares the $RCS$ from the ceilometer in a particle-free region with the molecular attenuated backscatter that can be calculated using Rayleigh theory. Computing the Rayleigh fit, by comparing the gradient with altitude (the slope) of both profiles, a region in the ceilometer profile that has the same trend than the expected molecular profile can be found. In this study, we select regions with a difference in gradients below 1%. Thus, in that region or reference height ($z_{ref}$), $C_L^*$ can be calculated:





$$C_L^* = \frac{RCS(z_{ref})}{\beta_m(z_{ref}) \cdot T_m^2(z_{ref}) \cdot T_p^2(z_{ref})} \tag{3}$$

At this reference height, the backscattering is only due to molecules. The transmittance due to molecules ($T_m$) can be easily

determined from Rayleigh theory but the transmittance due to particles ($T_p$) is unknown. Thus, we define $C_L(z_{ref}) = C_L^* \cdot T_p^2(z_{ref})$ or, in a different way

$$C_L = \frac{RCS(z_{ref})}{\beta_m(z_{ref}) \cdot T_m^2(z_{ref})} \tag{4}$$

If this method is applied on days with a low aerosol load, where the transmittance due to particles is close to 1, we can consider $C_L(z_{ref})$ a good approximation for $C_L^*$. Thus, the total attenuated backscatter can be calculated applying the following equation:

$$\beta_{att}(z) = \frac{RCS(z)}{C_L(z_{ref})} \tag{5}$$

On the other hand, if we are able to find a reference height, in principle it is possible to apply the Klett-Fernald inversion algorithm (Klett, 1981, 1985; Fernald et al., 1972; Fernald, 1984) and obtain particle backscatter coefficient profiles. Heese et al. (2010) and Wiegner et al. (2012; 2014) showed the capabilities of ceilometers applying this inversion algorithm studying a few cases. When trying to automate this process, there are two main problems: (1) $z_{ref}$ must be a particle free

region and, due to the low signal to noise ratio, finding $z_{ref}$ is not always possible, or the region might be a non-particle free region that, on average, follows the molecular trend (they have a similar gradient). Also, we might find several regions that meet the criteria, but it is complicated to discriminate automatically which one is the most appropriate. And (2) finding the correct particle lidar ratio ($Lr$). The second problem is small at the wavelength of the ceilometers used in this study (1064nm), due to the relatively low sensibility of particle backscatter coefficient to changes in particle lidar ratio at infrared

wavelengths. However, it does have an impact on the particle extinction coefficient profiles.

The first problem can be solved by continuously calculating $C_L$. After finding $z_{ref}$ for the Klett-Fernald inversion, in this region, the ratio of $RCS(z_{ref})$ and $\beta_m(z_{ref})$ should be close to the calibration factor continuously calculated with Eq. (4).

Once the first problem is solved, for the second one, independent AOD measurement from AERONET sun-sky photometers can be used to adjust the Lr in order to match the integral of the particle extinction coefficient profile (i.e. particle backscatter

coefficient profile multiplied by the Lr) with the AOD. Wiegner et al. (2012) applied this procedure to a Jenoptik CHM15kx ceilometer obtaining reasonable values for the Lr.





These two processes can be combined and automated for the continuous calibration of ceilometer data. If AOD data is available, the procedure follows the next steps:

- First, temporal averaging of the profiles is performed. In this study 30-min profiles are used for comparison with lidar profiles.

- Second, for each profile a set of potential $z_{ref}$ is calculated comparing the profiles of the $RCS$ and $\beta_m$, which is obtained from a standard atmosphere profile scaled to ground temperature and pressure. The slopes are calculated over a 990 m window. All regions with slope differences below 1% are selected.

- For each $z_{ref}$, and Lr from 20 to 80 sr, Klett-Fernald inversion is applied and the resulting profile is integrated and multiplied by the Lr and compared to the AOD. The pair $z_{ref}$ and Lr that minimizes the difference with the AOD is
selected.

- Finally, $C_L$ is calculated using Eq. (4).

$C_L$ calculated with this method has the influence of the transmittance and will differ from the actual instrumental $C_L^*$ from Eq. (1) or Eq. (3). If $C_L$ obtained over a long period is restricted to small AOD values, this $C_L$ would approximate the actual $C_L^*$ and can be used in order to calculate the total attenuated backscatter (Eq. 5). On the other hand, this variability of $C_L$ with

$T_p$ can help discriminating if an inversion is correct in the absence of AOD data. Thus, after finding a $Z_{ref}$ and applying the Klett-Fernald inversion to a profile (without temporal coincident AOD data) the calibration factor for this profile (applying Eq. 4) has to be close to the $C_L$ calculated around that time constrained with AOD data.

The next section quantifies the differences between these inversions and the inversions calculated independently with a multi-wavelength Raman lidar.

**3.1 Lidar – ceilometer comparison**

During the dust outbreak affecting the Iberian Peninsula between 20 – 24 February 2016, the multi-wavelength lidar operated on 22 February between 7:30 to 14:00 UTC and on 23 February between 8:00 and 13:30 UTC. Elastic inversion using Klett – Fernald were applied to 30-min average profiles at 1064 nm using a fixed lidar ratio of 50 sr. Thus, a total of 24 particle backscatter coefficient profiles were obtained. Coherence of the inversion at 1064 nm was checked against Klett –

Fernald and Raman methods at 355 and 532 nm. The resolution of the multi-wavelength lidar (7.5 meters) has been downscaled to 15 meters for the comparison with the ceilometer.

The ceilometer elastic inversion, using Klett – Fernald method, was also applied to 30-min average profiles for the same period; a total of 15 profiles were successfully inverted. The calibration factor at the reference height was calculated. If negative $C_L$ are discarded, a total of 11 profiles are comparable with lidar inversions.

The $C_L$ value calculated using AOD as a constraint during the dust event period has a median value of $(8 \pm 5) \cdot 10^{10}$ m³ sr. This value is verified with respect to statistical parameters of the profiles to find out a value that indicates the goodness of the profile in comparison with the lidar profile. The normalized mean bias (NMB) in particle backscatter of the ceilometer



and lidar profile is calculated following Eq. (6). The center of mass of the profiles is calculated with Eq. (7), and also the relative difference between ceilometer and lidar center of mass. Finally, the coefficient of correlation (R) of the profiles is determined.

$$NMB = \frac{\overline{\beta_{ceil}} - \overline{\beta_{lidar}}}{\overline{\beta_{lidar}}} \qquad (6)$$

$$C_{mass} = \frac{\int_{z_{min}}^{z_{max}} z \cdot \beta(z)\,dz}{\int_{z_{min}}^{z_{max}} \beta(z)\,dz} \qquad (7)$$

In Eq. (6), $\overline{\beta_{ceil}}$ and $\overline{\beta_{lidar}}$ are the mean particle backscatter coefficient from ceilometer and lidar respectively for the entire retrieved profile, and $\beta$ in Eq. (7) may refer to ceilometer or lidar particle backscatter coefficient depending on the case.

Figure 2 shows for the 11 comparable profiles, the calibration factor at the reference height on the ceilometer profile versus the NMB (top panel), center of mass relative differences (middle panel) and R (bottom panel). Ceilometer profiles with a calibration factor closer to the median calibration factor have inversions closer to the lidar inversions. Figure 2 also shows that, for the NMB and R, the difference between the calibration factor and the median calibration factor are related, and the farther the profile calibration factor is from the median value, the worse the statistic value. Thus, considering a threshold, the

quality of the profiles can be automatically determined. Considering a 33% around the median value of the calibration factor (half the standard deviation), we obtain four profiles that have a NMB smaller than 15%; the center of mass of the profiles is practically the same, with a maximum difference of 3% and finally, R of the profiles are above 0.92.

A sequence of ceilometer and lidar particle backscatter profiles from 23 February 2016 is shown on Fig. 3. The first ceilometer profile (marked in blue) has a calibration factor of $1.18 \cdot 10^{11}$ (m³ sr) and hence is rejected according to the

threshold described above. For this case, the NMB of the ceilometer and lidar profiles is -0.31, the center of mass relative difference is -0.06 and the R is 0.84. The other four ceilometer profiles (marked in red) have calibration factors within the 33% of the median calibration factor. The profiles on 23 February at 9:00 and 9:30 UTC correspond with a decoupled dust layer. Those profiles have a NMB of -0.08 and 0.1 respectively, the center of mass relative difference is -0.01 and -0.03 respectively, and R is 0.95 and 0.97 respectively. The profiles on 23 February at 12:00 and 12:30 UTC show that the

previous dust layer is mixed with the boundary layer. In these cases, profiles have a NMB of 0.14 and -0.12 respectively, the center of mass relative difference is 0.006 and -0.01 respectively, and R is 0.99 and 0.93 respectively.

## 4. Results

The capabilities of the ceilometer network for aerosol optical properties characterization and the near real-time processing have been tested with the analysis of the African dust outbreak that affected the Iberian Peninsula on 20 February 2016 and

persisted until 24 February 2016.

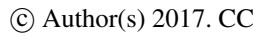



## 4.1 Description of the dust episode

The evolution of the dust outbreak is illustrated in Fig. 4 where a sequence of the false-color RGB dust images from MSG-SEVIRI is shown. This product makes use of three thermal satellite channels to contrast the brightness temperature signal between surface, cloud and dust (Lensky and Rosenfeld, 2008), in a color scheme in which dust appears in magenta. The

5 presence of dust plumes over the High Plateau located between the Saharan and Tell Atlas in Algeria at 12:00 UTC on 20 February 2016 is shown in Fig. 4a. Dust migrated NW and passed over the Alboran Sea from the Algerian-Moroccan border around 14:00 UTC, reaching the southern Iberian Peninsula at 18:00 UTC (Fig. 4b) and continued moving north-westward (Fig. 4c). A second dust plume migrated northwards on 21 February 2016 at 16:00 UTC (Fig. 4d).

SYNOP meteorological observations and aerodrome routine (METAR) and special (SPECI) reports in Northern Africa

recorded a strong reduction in horizontal visibility (MOR, Metorological Optical Range), down to 2 km, between 7:00 and 8:00 UTC (20 February) at distant locations situated at the edges of the Great Western Erg in Algeria. At the eastern part of the Western Erg and in the Great Eastern Erg, visibility lowered to less than 5 km between 9 and 11:30 UTC. In good accordance with the satellite images, at the Saharan Atlas and the High Plateau area, with heights over 1000 m, visibility less than 2 km were recorded at 10:00 UTC at Mecheria, while at the other stations in that area values went down to 2-3 km at

12:00 UTC. High relative humidity and clouds were found in the western most sites, also in agreement with the satellite images. It is remarkable that no significant visibility reduction was reported at the north-faced downslope areas of the Tell Atlas and in the Rif mountains, close or at the coast of Algeria and Morocco. This indicates that dust was uplifted before passing over the northern slope of the Atlas range and the North African coast. Correspondingly, no station in the southern Iberian Peninsula reported a reduction in visibility when the dust plume reached Spain.

The entrance of dust-laden air masses above the ground level in the Iberian Peninsula is confirmed by a back-trajectory analysis performed with HYSPLIT (Hybrid Single-Particle Lagrangian Integrated Trajectory) model (Draxler and Hess, 1998; Stein et al, 2015) using ERA-Interim data of 0.5º resolution. The trajectory analysis provides an estimate of the range of heights at which the dust-laden air masses passed over the study sites. This is illustrated in Fig. 5 for two stations: one on the south where dust reached the Iberian Peninsula (Granada); and in the westernmost station, on the direction of the dust

mobilization (Badajoz). The height vs latitude plots show that the dust plumes reaching Granada on February 20, 18:00 UTC and Badajoz on February 21, 00:00 UTC, arrived at mid-levels in the lower troposphere after being uplifted in the southern slope of the Saharan Atlas from heights between 500 and 1200 m above the ground in that area (black trajectory on Fig. 5). Those trajectories end at around 3250 and 2500 m asl in Granada and Badajoz respectively. Figure 5 also shows that the trajectories reaching Granada at 2250 m asl passed over Africa before the dust mobilization took place and the ones below

had no African origin. Finally, the trajectory reaching Granada at 4250 m asl followed the mid-upper level tropospheric circulations. In Badajoz, results are analogous: at 1000 m asl and below trajectories arrived from the north, between 1250 m asl and 2000 m asl they arrived from Africa but the air parcels were located north of the area where dust was observed in the



morning of February 20. The air parcels reaching Badajoz between 2250 m asl and 3250 m asl were previously located in the area where dust was being observed, while at upper levels trajectories followed the mid-upper circulation pattern.

## 4.2 Synoptic scenario and context

Several atmospheric features at mid-upper levels were relevant for this episode (Fig. 6), as they promoted instability near the
surface and induced the dust transport in the lower free troposphere: (1) the amplification and break up of a Rossby wave in the eastern Atlantic resulted in a trough that became isolated as a cut-off low over the Atlas Mountains on 19 February. A shallow cyclone was then originated leeward of the Atlas. From early 21 February, the cut-off low displaced off the Moroccan coast and centered southwest of St. Vincent Cape. On 22 February it decayed bringing the Iberian Peninsula under the influence of the Azores and North African subtropical highs, with dominant zonal flows. (2) An upper-level anticyclone
over a wide area centered over Niger-Chad, which during the episode intensified and extended northwards to the western Mediterranean. This high-pressure influenced circulation at mid-upper levels in combination with the cut-off low. (3) Moisture flux at mid tropospheric levels, which entered from the central Atlantic into the African continent below 20ºN and was transported to northern Africa (at 400-550 hPa according to the radiosoundings in the area) between the upper-level trough and the high-pressure system. The tropical air masses are well appreciated in the satellite imagery as an elongated
cloud band moving north and eastward, and so are the convective clouds formed ahead of the band. The tropical-extratropical interaction between the advected tropical moisture and the upper-level trough located over the Atlas range is linked to convective precipitation in northwestern Africa, see Knippertz (2003) and references therein; it is more frequent in summertime, however. Although not mentioned in these works, a reduction in visibility due to dust mobilization over Algeria can also be observed when revisiting their case studies. Divergence at upper levels (250 hPa) and low-level (850
20  hPa) convergence are found over the area where the gust front mobilized the dust on 20 February. The interaction with the Ahaggar Mountains in southern Algeria possibly enhanced convection and low-level instability. Convective precipitation was registered at several locations of eastern Spain when the cloud band passed over the area in the second half of the episode. From 22 February onwards the cloud band and local convective situations were gradually displaced to the Mediterranean, as zonal flows began to dominate.
At low levels, the low-pressure formed in the lee of the Atlas moved to the SW of the St. Vincent Cape on 21 February following the upper level instability. The low was then intensified and influenced northern African and most of the Iberian Peninsula. In addition, high pressures over the western Mediterranean were formed when the Rossby wave train progressed to the east and retreated poleward. Then, the North African high, which was previously located over Libya at 850 hPa, extended to Tunisia and Algeria and was gradually intensified in connection with the northward extension of the high-
pressures at upper levels, which arrived (along with the cloud band on its western flank) to the western Mediterranean Basin. The advection of dust-laden air masses to the Iberian Peninsula was driven by both the low located to the SW of the Iberian Peninsula and the North African high. The presence of these two synoptic systems corresponds to one of the typical synoptic situations leading to dust transport over the Iberian Peninsula (Rodriguez et al., 2003; Escudero et al., 2005). During the





episode, however, two distinct strong plumes were transported from northern Africa to the Iberian Peninsula in consecutive days and showed a different evolution. Dust mobilized by the gust front on 20 February south of the Saharan Atlas and north of the Ahaggar migrated west and northward to the Iberian Peninsula, as showed in the satellite images, forming a curve-shaped plume over Iberia due to the cyclonic shear imposed by the low. The second strong dust plume was mobilized and

transported northwards on 21 February on the western side of the North African high, driven by the intensification of this high pressure system, which was the dominating feature in the second half of the episode. In this second case, dust was advected mostly below the cloud band and affected the eastern part of the Iberian Peninsula as well as most of the western Mediterranean basin.

The low pressure system weakened on early 22 February and the region was increasingly dominated by the Azores and the

North African highs. As a consequence, zonal flow swept the first dust plume along northern Spain from west to east in subsequent days. The second dust plume, which was moving northward along eastern Iberia, was also displaced to the Mediterranean. The study region was then under high pressures and the event ended.

## 4.3 Ceilometer data analysis

The vertical structure of the dust event described above has been characterized with the Iberian Ceilometer Network

(ICENET). Firstly, by applying the calibration factor and obtaining the attenuated backscatter profiles and, secondly, by applying the inversion and obtaining particle backscatter profiles. In addition, the integral of the backscatter profiles multiplied by the lidar ratio is used to estimate the AOD during the event and the center of mass of the backscatter profiles is considered as an indicator of the presence of a decoupled aerosol layer (a dust plume in this case) or the entrainment of the aerosol layer into the boundary layer. All these products were calculated in near real-time and serve as an example of the

promising capabilities for real-time characterization of singular events with a network of distributed ceilometers.

Figure 7 shows time series of total attenuated backscatter profiles, i.e. calibrated profiles, for the five ceilometers used in this study. From top to bottom the series correspond to Granada, Tabernas, Badajoz, Valladolid and Montsec stations respectively. Tabernas station is covered by clouds during most of the event and Montsec station is also affected by clouds during part of the event.

Dust arrives first at the stations in Granada and Tabernas (on 20 February at 18:00 UTC). As the dust plume moves northwestward we observe the dust plume in Badajoz (on 21 February at 00:00 UTC), and Valladolid (on 21 February at 06:00 UTC). At Montsec, the dust plume is detected on 21 February at 12:00 UTC). The second plume brings the cloud band and this is visible at Tabernas station around 12:00 UTC on February 21 and a bit later at 21:00 UTC on 21 February at Montsec station. Finally, the displacement of the dust from west to east at the end of the event, when the cut-off low

weakens appears as a dust plume at Valladolid on 22 February at 15:00 UTC, at Badajoz station on 22 February at 21:00 UTC, and at Granada station on 23 February at 06:00 UTC. Tabernas and Montsec are influenced by the second dust plume and the cloud band and this is not as clearly visible as on the other stations. Another feature that is observed in Fig. 7 is that the dust plumes, specially the first one, are entrained into the boundary layer rapidly.



After applying the inversion, a quantitative comparison of stations is possible as shown in Fig. 8 for different stages of the dust outbreak. The beginning of the outbreak, when the first plume arrives at the different stations is shown in Fig. 8a. The center of mass of the dust plumes is about 3000 m asl for all stations. Additionally, for Granada and Badajoz, we observe that the height of the peak in particle backscatter coefficient is in accordance with the backward trajectory analysis shown on

Section 4.1. The arrival of the second plume is shown in Fig. 8b for all sites on 22 February at 06:00 UTC. At this stage, we observe that Granada and Tabernas stations (which are only 100 km apart) show very different behavior in particle backscatter and also in the height of the dust plume. Finally, Fig. 8c shows the final part of the outbreak when dust is mobilized eastwards to the Mediterranean Sea. In this case, dust is below 2000 m asl for Granada and Tabernas, whereas for the rest of the stations it is still observed above 3000 m asl. In general, the particles backscatter coefficient profiles indicate a

stronger intensity of the event in this stage of the event, after the second plume arrives specially for Granada and Tabernas.

For the entire dust outbreak period and all stations the integral of the backscatter profiles is shown in Fig. 9a. This parameter allows identifying the beginning of the dust event for each station. Thus, it is observed and increase of the integral of the backscatter in Granada around 20 February at 19:30 UTC, in Badajoz is detected around 21 February at 5:30 UTC, in Valladolid at 16:30 UTC and in Montsec at 17:00 UTC. Due to clouds, this increase of the integral of the backscatter is not

observed in Tabernas. The influence of the dust load after the first plume masks the arrival of the second plume, but the dust mobilization towards the Mediterranean sea is observed again at Badajoz (around 22 February at 20:00 UTC) and in Montsec at 23:00 UTC. The change of the integral of backscatter to larger values is in coincidence with the starting time observed in the total attenuated backscatter temporal series and it is in accordance with the satellite observations and back trajectory analysis. Additionally, the center of mass of the particle backscatter coefficient profiles is used to monitor the

evolution of the profile region with more predominance of aerosol particles. Thus, in Fig. 9b for Granada before the event, the center of mass is about 1500 m asl, and when the dust arrives the center of mass is elevated to 2500 m. After 9 hours the center of mass is about the same as before the event, indicating that, possibly, the dust plume is no longer decoupled, and it is entrained into the boundary layer. A similar behavior is observed for Badajoz, Valladolid and Montsec stations. Again, the second plume is not observed in changes of the center of mass but the mobilization of dust towards the Mediterranean Sea is

observed as an increase of the center of mass of the profiles for Badajoz, Valladolid and Tabernas.

## 5 Conclusions

The use of ceilometers for the characterization of optical aerosol properties is possible but, due to the weak signal, it is important to screen out profiles in order to ensure the quality of the inversion. In addition, due to the vast amount of data, it is important to perform all these operations in an automated, unsupervised way and, preferably, in near real-time. The

methodology proposed uses ancillary data from sun photometer in order to constraint the calibration of the ceilometers. The time series of this calibration is used to determine the quality of the inversions selecting those that present, at the reference height, a ratio of the backscattering signal to molecular attenuated backscatter within the 33% of a median calibration factor





for the same period. A comparison with independent lidar measurements indicates that this method allows the automatic discrimination of the quality of the inversions with ceilometers. During this comparison a difference of 15% in backscatter coefficient is observed. Thus, it is feasible to routinely provide particle backscatter coefficient profiles with ceilometers.

The inverted profiles obtained with ceilometers could be used for elevated aerosol layer alert by setting a threshold on the
5 particle backscatter coefficient values of the profile and are potentially useful for model assimilation and evaluation since all the processing is automated and in near-real time.

This method has been applied to a group of ceilometers (the Iberian Ceilometer Network, ICENET) and tested during a dust outbreak reaching Spain on 20 February 2016 until 24 February 2016. This dust event affected all ICENET stations with two distinct plumes reaching the Iberian Peninsula following different paths and a final stage where zonal flows swept the dust
towards the Mediterranean Sea. This scheme of dust mobilization is unusual for this season of the year and the intensity, spatial coverage and duration of the event makes it perfect as a test for monitoring purposes with the ceilometer network. The calibration of the ceilometers allows a qualitative monitoring of the event while the inversions provide quantitative information. Thus, ceilometers can complement lidar stations that, in principle would operate intermittently and with less spatial density. It is worth noting that it has been observed differences on profiles 100 km apart. This reinforces the need of
providing vertical profiles of aerosol optical properties with a dense spatial resolution.

Parameters extracted from the particle backscatter coefficient profiles such as the integral or the center of mass can also give a quantitative idea of the presence of an elevated aerosol layer. These parameters are expected to increase with an elevated aerosol layer and the second one can be used as a rough indicator for the deposition velocity of an elevated aerosol layer by comparing a time series of these values.

**Acknowledgments**

This work was supported by the Spanish Ministry of Economy and Competitiveness through projects CGL2012-39623-C02-01, CGL2013-45410-R, CGL2014-56255-C2-1-R, CMT2015-66742-R, CGL2015-70741-R, CGL2015-73250-JIN and CGL2016-81092-R, by the Andalusia Regional Government through project P12-RNM-2409, by the Castilla y Leon Regional Government through project VA100U14, by the Junta de Extremadura (Ayuda a Grupos de Investigación
GR15137) and by the European Union's Horizon 2020 research and innovation program through project ACTRIS-2 (grant agreement No 654109). The authors thankfully acknowledge the FEDER program for the instrumentation used in this work. This work was also partially funded by the University of Granada through the contract "Plan Propio. Programa 9. Convocatoria 2013". Marco Pandolfi is funded by a Ramón y Cajal Fellowship (RYC-2013-14036) awarded by the Spanish Ministry of Economy and Competitiveness. The authors would like to acknowledge the valuable contribution of the
discussions in the working group meetings organized by COST Action ES1303 (TOPROF).





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





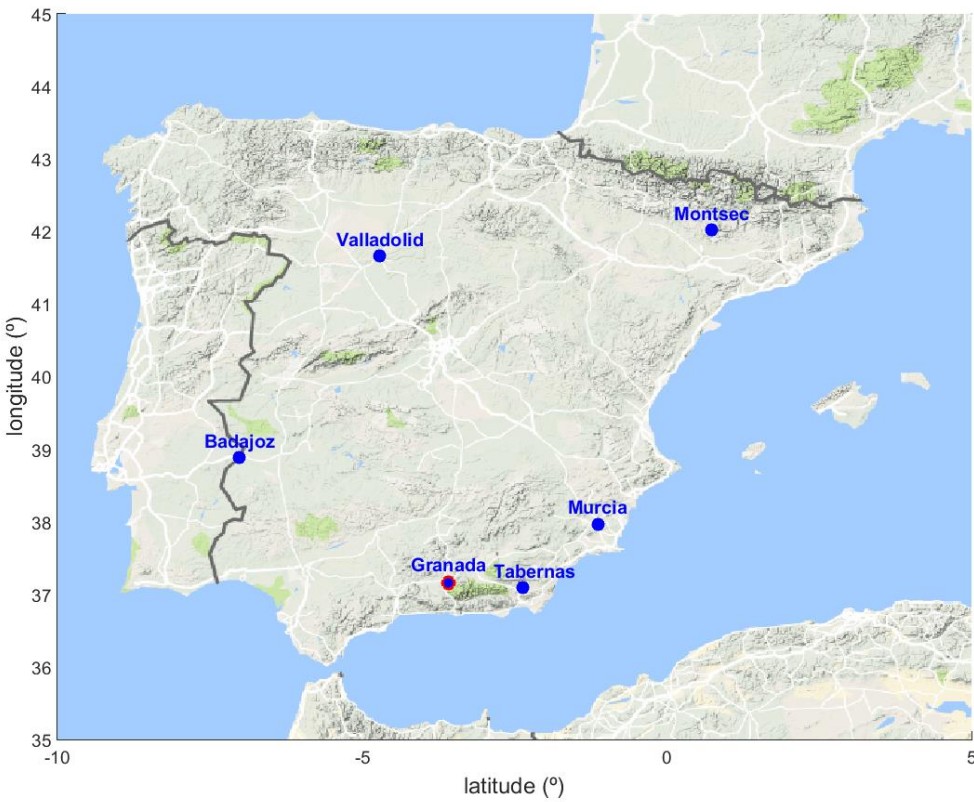

**Figure 1. Map of the Iberian Peninsula showing the location of the ceilometers. In Granada station (circled in red) a co-located multiwavelength raman lidar is also available.**




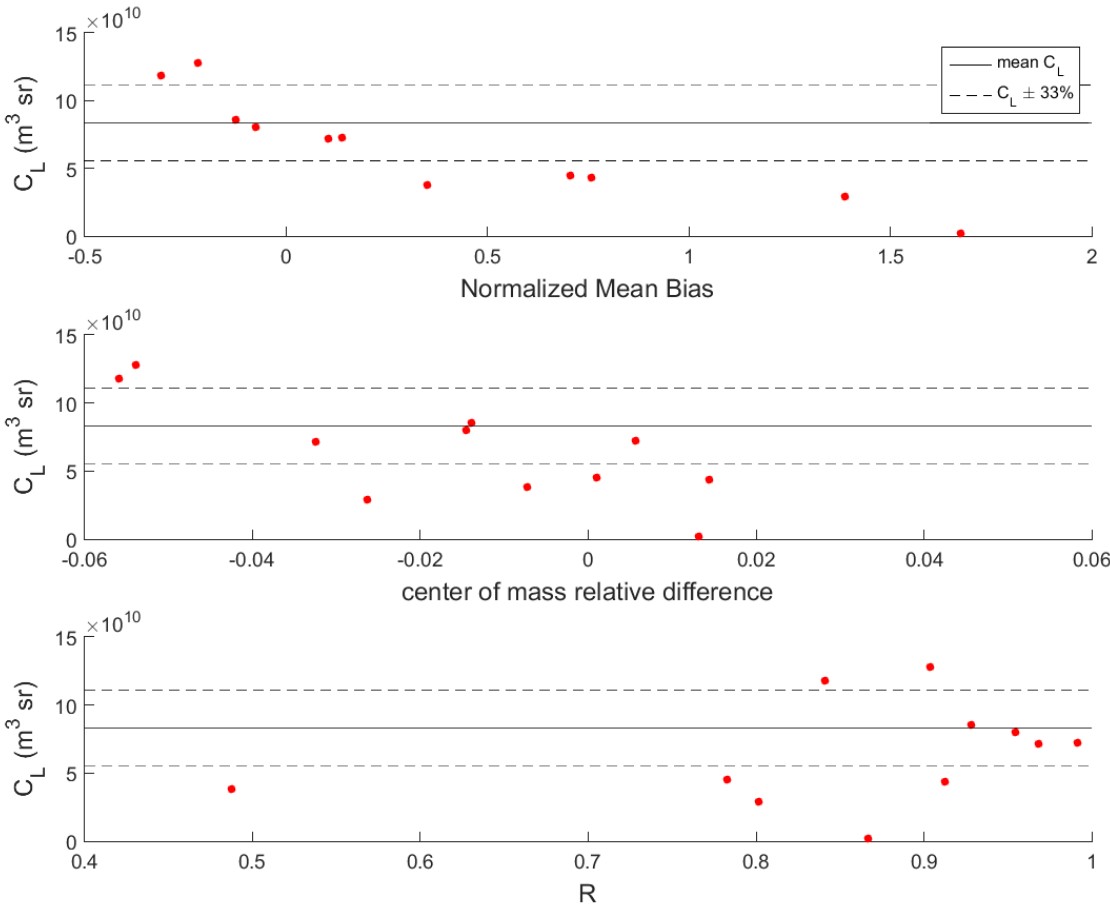

**Figure 2. Ceilometer Calibration Factor ($C_L$) vs. normalized mean bias (NMB) on top panel, relative difference in center of mass on middle panel and coefficient of correlation (R) on bottom panel. Horizontal line indicates the mean $C_L$ for the dust event period and dashed lines indicates the 33% around this mean value.**




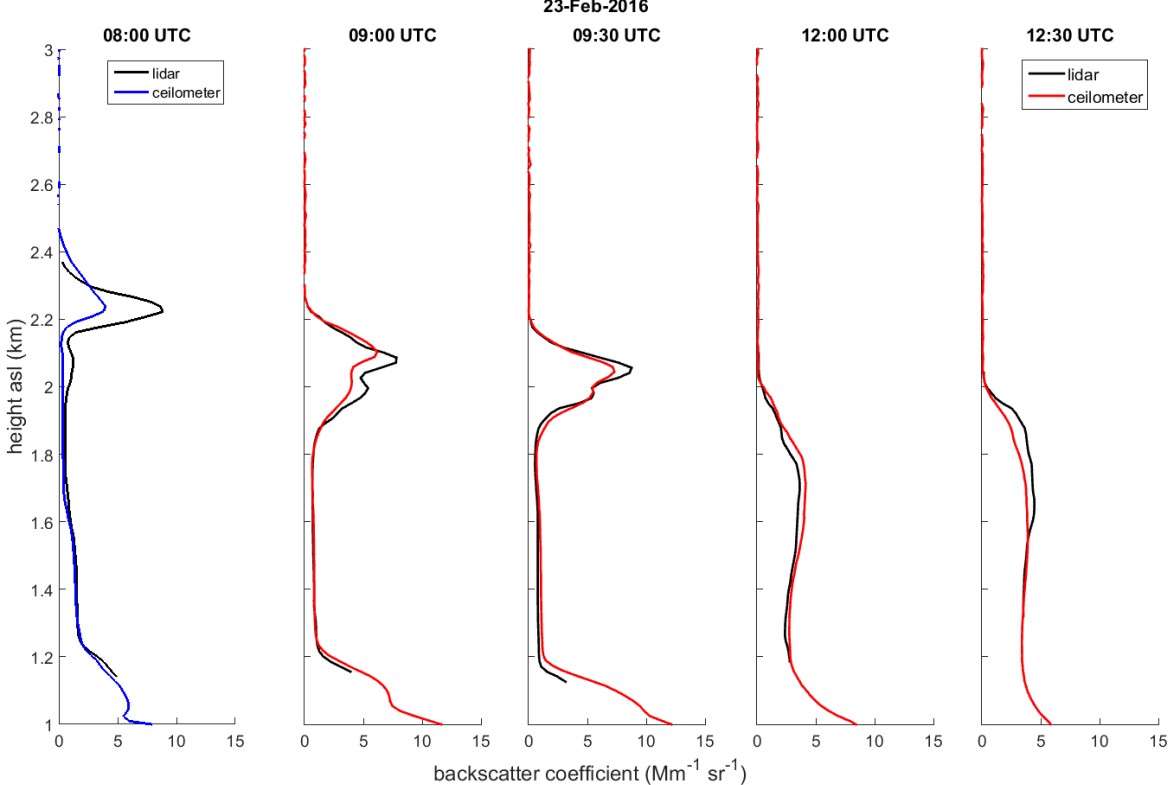

**Figure 3. Lidar and ceilometer particle backscatter profiles for five cases on 23 February 2016. The first case (marked in blue) is a rejected ceilometer profile and the other four cases (marked in red), are cases with a ceilometer calibration factor within the 33% of the median calibration factor. Errors on lidar profiles have been estimated with a Montecarlo technique (Pappalardo et al., 2004; Mattis et al., 2016) and are in the order of $10^{-9}$ ($m^{-1}$ $sr^{-1}$), therefore they are not shown.**



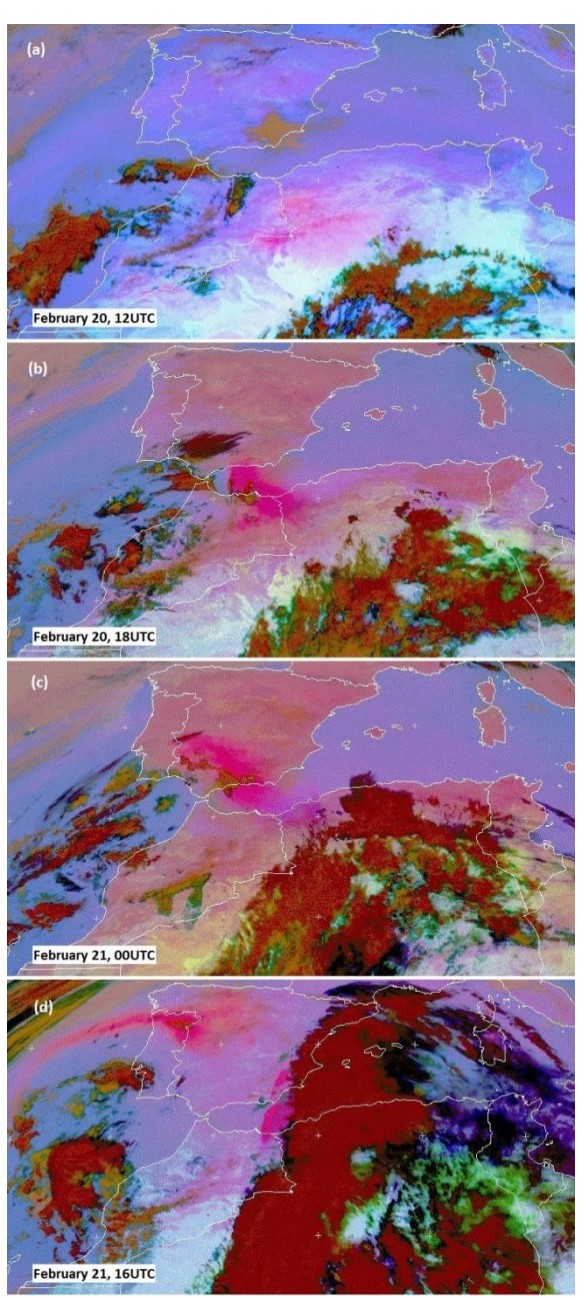

**Figure 4. False-color RGB dust image from MSG-SEVIRI showing different stages of the dust outbreak (dust appears magenta).**





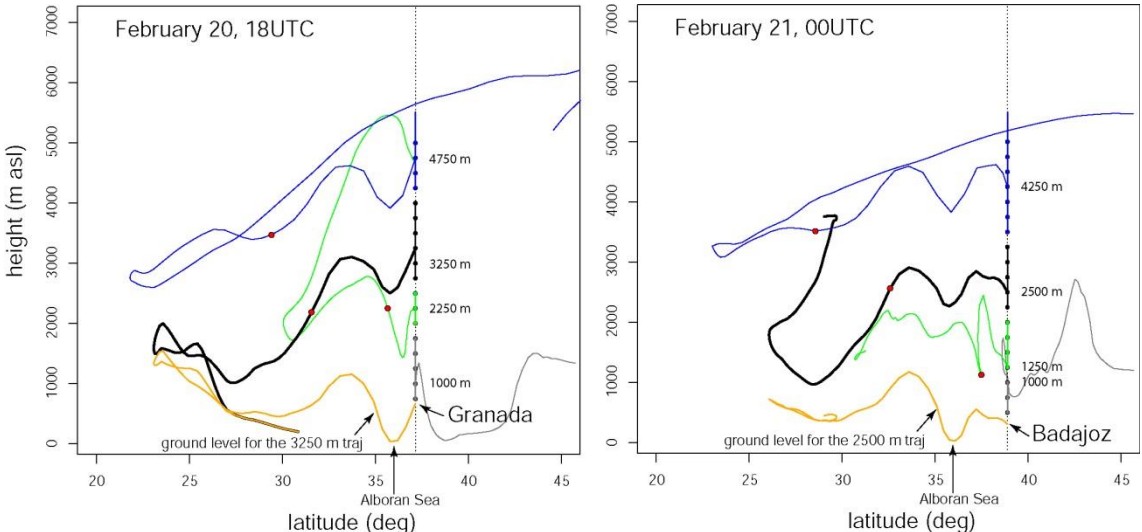

**Figure 5. Evolution of the air parcels reaching Granada on February 20, 18:00 UTC (left) and Badajoz on February 21, 00:00 UTC (right) at different heights. Back-trajectories were calculated from the ground level to 5000 m asl, at every 250 m. Lines in grey indicate trajectories arriving at the lowest levels, with no African history; in green are trajectories that passed over the southern slope of the Saharan Atlas before the observed dust mobilization; in black are the trajectories followed by the parcels residing at the times and area where dust was observed; the trajectories residing at higher levels are depicted in blue. One representative trajectory is shown for each evolution and the altitude interval is shown in the same color as the representative trajectory. The brown line corresponds to the ground level for the trajectories more associated to the dust advection (thick black lines). The location of the air parcels around the time of observation of the dust plumes is shown as a red circle.**





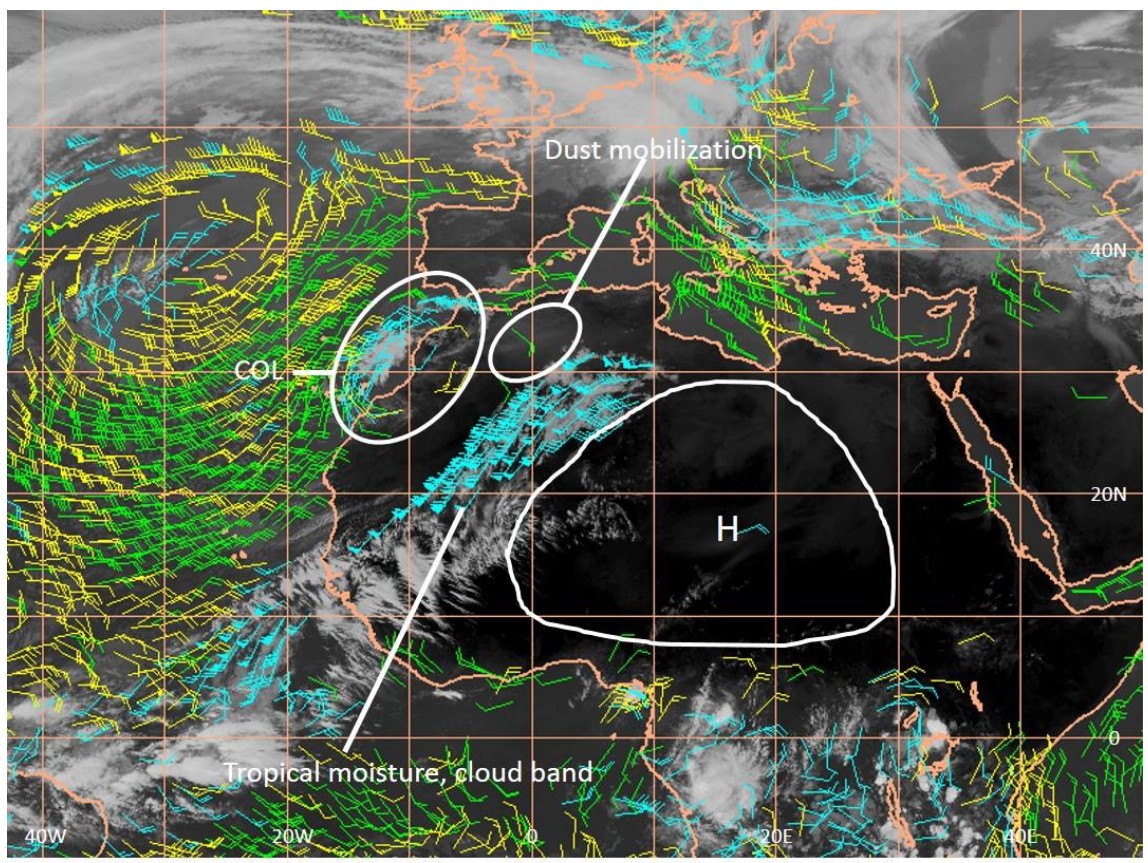

**Figure 6. Relevant mid/upper -level features for the African dust outbreak: cut-off low (COL), upper-level anticyclone (H) and the tropical moisture intrusion on February 20, 12 UTC (image from University of Wisconsin-CIMSS).**







**Figure 7. Ceilometer time series of total attenuated backscatter representing the evolution of the dust outbreak between 20 and 24 of February 2016 (the color scale is logarithmic).**

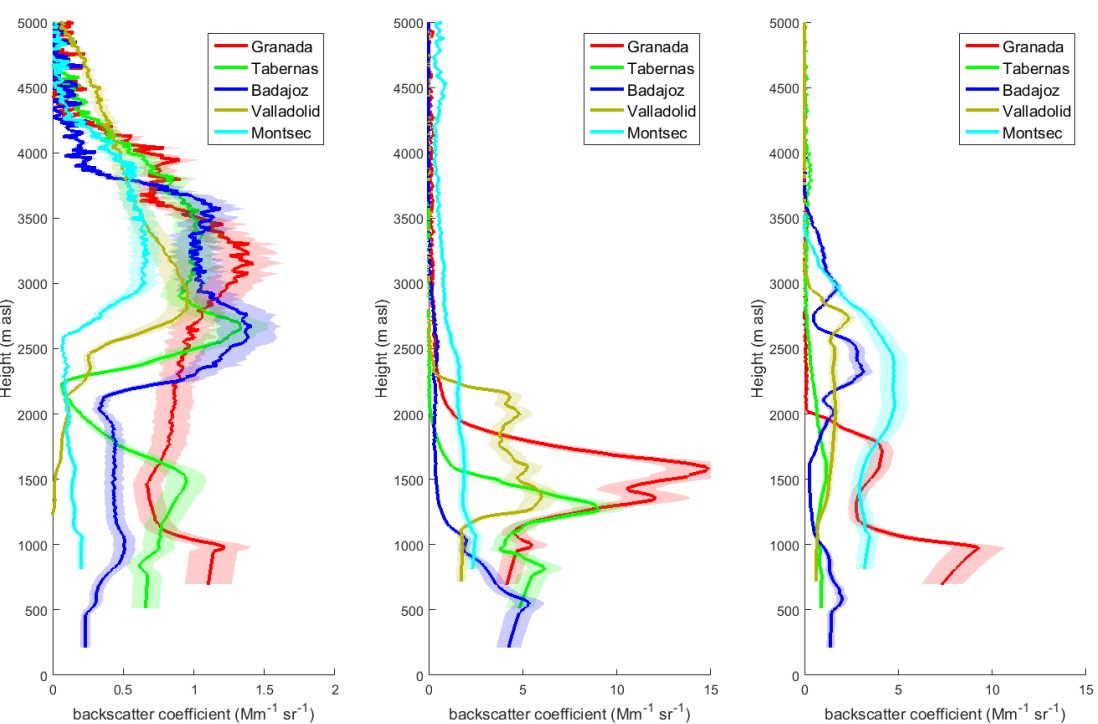

5    **Figure 8. Particle backscatter coefficient profiles for all stations at the beginning (a), middle (b) and final stage (c) of the outbreak (note that x axis has different scale and the profiles start at ground level). Shaded area represents the 15% uncertainty.**

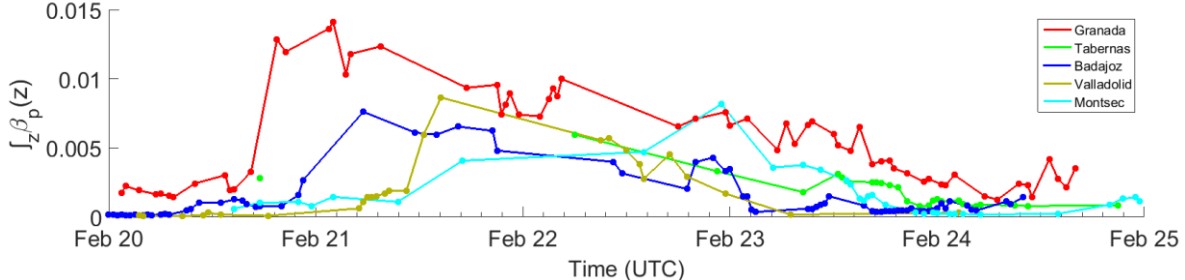



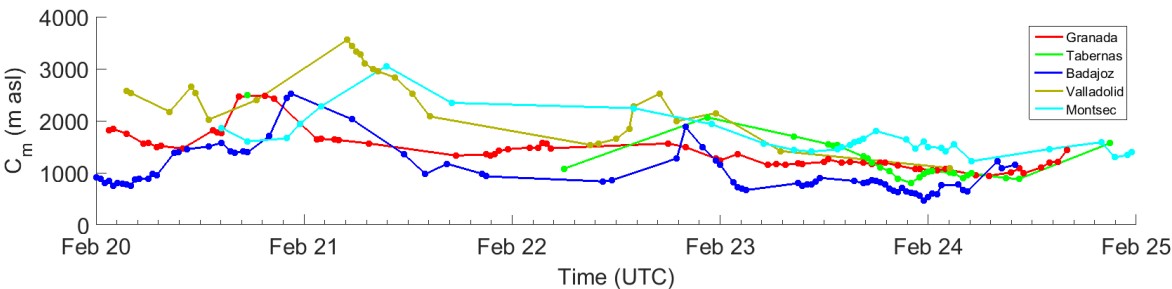

**Figure 9. Time series of the integral of the particle backscatter coefficient for all stations (top panel) and time series of the center of mass of the backscatter profiles for all stations (bottom panel).**

5    **Table 1. Description of the Iberian ceilometer network sites.**

| Site (code) | Managed by | Location (Latº/Lonº) | Height (m asl) | Additional Instruments |
|---|---|---|---|---|
| Granada (UGR) | Atmospheric Physics Group. University of Granada. | 37.16ºN/3.58ºW | 680 | CIMEL CE 318 (co-located) Multi-Wawelength lidar (colocated) |
| PlataformaSolar de Almería -Tabernas (PSA) | Institute of Solar Research. German Aerospace Center. | 37.09ºN/2.36ºW | 500 | CIMEL CE 318 (co-located) |
| Badajoz (UEX) | AIRE Group. University of Extremadura. | 38.88ºN/7.01ºW | 199 | CIMEL CE 318 (co-located) |
| Valladolid (UVA) | Atmospheric Optics Group, University of Valladolid. | 41.66ºN/4.71ºW | 705 | CIMEL CE 318 (co-located) |
| Montsec (MSA) | Institute of Environmental Assessment and Water Research. Spanish Research Council. | 42.02ºN/0.74ºE | 800 | CIMEL CE 318 (42.05ºN/0.73ºE/1570m amsl) |
| Murcia (UMH) | Statistical and Computational Physics | 39.98ºN/1.13ºW | 69 | CIMEL CE 318 (co-located) |





| | | | |
|---|---|---|---|
| | Lab, Miguel Hernández University. | | |