# Peer review of "Near real time processing of ceilometer network assisted with sunphotometer data: monitoring a dust outbreak over the Iberian Peninsula"

_Atmospheric Chemistry and Physics, 2017_

## Referee Comment (RC1) · Anonymous Referee #2 · 22 Jun 2017

The manuscript entitld "Near real time processing of ceilometer network data: characterizing an extraordinary dust outbreak over the Iberian Peninsula" by Alberto Cazorla, Juan Andrés Casquero-Vera, Roberto Román, Juan Luis Guerrero-Rascado, Carlos Toledano, Victoria E. Cachorro, José Antonio G. Orza, María Luisa Cancillo, Antonio Serrano, Gloria Titos, Marco Pandolfi, Andres Alastuey, Natalie Hanrieder, and Lucas Alados-Arboledas (doi:10.5194/acp-2017-151) is appropriate for publication in Atmos. Chem. Phys.

The authors describe a new method for the combined processing of ceilometer data and sun photometer observations and they use the data obtained from this procedure

for the characterization of a dust event over the Iberian Peninsula. The development of methods for quantitative retrievals of aerosol profiles from ceilometer data is relevant to the scientific community. Nevertheless, the manuscript needs significant improvement.

The most critical points of the manuscript which should be improved before final publication are:

1) The title does not clearly reflect the content of the paper. The input of sun photometer data is essential for the proposed method, but the title does not provide any hint that the described method is not usable for standalone ceilometer instruments. Further, the manuscript provides no proof, why the described dust outbreak event is extraordinary.

2) What makes the described event extraordinary? Is it the meteorological situation, intensity, duration or something else? If the event was extraordinary in a certain aspect, this statement should be proved by comparison with typical events. If the extraordinariness of the event cannot be corroborated, the title should be adopted.

3) Why is the transmittance due to particles neglected? The authors describe how sun photometer data are used to constrain the Klett-Fernald inversions. Thus, all relevant information for the calculation of Tp is available. Why is it not used?

4) The authors shall provide an estimation of the difference between CL* and CL for different particle optical depths.

5) It would be nice to have an example plot for illustrating the calibration method which is described at page 7. It would be even better to show two examples, one of a clear day and one of a polluted day.

6) The manuscript suffers from a major internal conflict. First, it is introduced that the calibration parameter CL can be retrieved only on days with low aerosol load. But in the next part, a period with very high aerosol load is used for demonstration and validation of the method. All CL values which are derived during the dust period will lead to a systematic bias of profiles if they are applied to measurements in clean conditions.

[Figure]

The retrieved CL values are systematically too small due to neglecting Tp. To overcome this problem, the authors should derive and present a longer time series of CL values, with clean periods before and after the dust event. According to the theory, only the maximum values of CL in this time series (clean periods) are ok. Those maximum values before and after the dust event should be used for the retrieval of the attenuated backscatter profiles during the event. The lidar profiles can be used for the validation of these ('clean') CL values. But, it makes no sense to derive CL values during the dust event (even if constrained with lidar profiles) without taken into account Tp.

All attempts to constrain CL values measured during the dust event should be removed from the manuscript. This includes Figures 2 and 3, and all text below equation 7 and the begin of section 4.

Besides, the use of the correlation coefficient and center of mass as measures of goodness of the calibration seems to be of little help. Even if the calibration value is wrong, both profiles should have the same shape (and therefore the same Cmass and high R) because they are measured under the same atmospheric conditions.

7) The link between the two parts of the manuscript (methodology and results) is week.

8) The description of the meteorological situation during the event is quite lengthy while some interesting measurement data are not provided, e.g. ,depolarization profiles from the lidar, time series of optical depth and Angstroem exponent, fine-to-coarse mode fraction etc. from the photometer network.

9) In general, the description of the event would be more useful if the authors could provide references to other studies about dust over the Iberian Peninsula. How often do events like the described dust outbreak occur at the Iberian Peninsula? What are typical pathways and meteorological conditions? What are typical optical properties (e.g. optical depth) of the dust? What are typical geometrical properties of the dust layers? What makes this event special compared to others?

Further major concerns are:

10) abstract, the last sentence "... quantitative optical aerosol characterization with ceilometers..." is misleading. Ceilometers can be used for the quantification of the aerosol layering or aerosol load in terms of particle backscatter coefficients, but they cannot be used for characterization. 'Characterization' is often used as synonym for 'detection of aerosol type' or 'retrieval of intensive optical properties (like lidar ratio or Angstroem exponent)'. Ceilometers cannot provide this kind of information.

11) p2, l22: It is strange to write about the inadequate quality of satellite products in a paper about aerosol profiles from ceilometers. Certainly, products from ceilometers are very useful, but usually they do not provide profiles of higher quality than space-borne lidars.

12) p2, l28: There are more relevant references, e.g. Flentje et al. 2010.

13) introduction: The order of sentences in the last two paragraphs seems to be somewhat randomly. Maybe due to copy-and-paste? An outline to the structure of the manuscript is missing.

14) equation 1: CL* is not a constant in a strict sense. It changes on long time scales, e.g. due to laser aging or window contamination. Better to use 'parameter' instead of 'constant'.

15) p7, l29: how can negative CL values be explained? NMB is usually calculated as mean value of the bias profile ( b_ceil(z) - b_lidar(z) ) / b_lidar(z)

16) Figure 5: This kind of trajectory plot seems to be less informative than the traditional plots (with a map projection and a time-altitude plot below). To which times of observation do the red dots refer to?

17) Figure 6: Is it really necessary to show this figure?

18) Figure 7: The labels of the color bar are unreadable. Time axes of the different

stations have different tick scales.

19) Figure 8: Where does the uncertainty of 15% comes from? Please, indicate the measurement times of the individual profiles by vertical lines in figure 7.

20) Figure 9: Please, add time series of AOD and columnar mean lidar-ratio (whenever Integrated bsc and AOD are available). Also time series of Angstroem exponents and fine-to-corase mode fractions would be interesting.

Minor / pure technical or language comments:

21) The readability of the text could be improved by splitting some long sentences into shorter ones, e.g.

* p1, l23+24

* p1, l25+26

* p3, l1+2

* p3, l18-20

* p5, l28-31

22) p1, l25: it is not clear to which method the term "this method" refers.

23) p1, l26: the date of the event is described several times in the manuscript as ... on 20 February and lasted until 24 February... -> it would be better to write "between... and .." or "... lasted from .. to..."

24) p1, l21 + p2 l1: what refers "their" to?

25) p2, l6: The terms "in-situ" and "surface measurements" are often used for ground based remote sensing instruments like lidars, in contrast to space-borne instruments. Maybe a term like "measurements of aerosol properties at ground level" could be used instead.

26) p2, l7: when speaking about a covered area, it would be better to use "Europe" instead of "European Union".

27) p2, l10: (and elsewhere in the paper): vertical resolved -> vertically resolved

28) p2, l31: make complicate -> hinder?

29) p3, l18-20: the calibration is used to validate the inversion?

30) p4, l2: comparable -> calibrated?

31) p4, l4: constraint -> constrain

32) p5, l16: not overlap -> no overlap

33) p5, l28: what means "computing the Rayleigh fit"? -> ... particle free regions are determined by comparing the gradient... If the difference is below 1%, we can assume particle free conditions...

34) p7, l12: ... has the influence of ... -> ...is influenced by...

35) p10, l26 ... northern African... -> ... northern Africa...

---

## Referee Comment (RC2) · Anonymous Referee #1 · 25 Jun 2017

The manuscript by Cazorla et al. is suited for publication in acp after some revisions. It shows a method how a network of ceilometers could be calibrated in an unattended and automatic way and the paper demonstrated the usefulness of such a network in a case study of desert dust over the Iberian Peninsula.

Major critical comments

I am a bit confused on the main objective of this manuscript. It tackles several issues and by doing so it creates confusion. First, the papers deals with near real time data processing, however it misses an evaluation how good the near real time results are in comparison to results which are obtained not in near real time. Second, the paper

describes an automatic calibration method for ceilometers. It highlights that the calibration works best for low AOD and then it is applied for a case of desert dust where AOD is not low. Third it describes quite lengthy the meteorological situation of the dust episode, describes the episode as it was observed by ceilometers but then it lacks the description of other measurements during this episode. I am confused what the main objective of this study is. The authors could re-formulate parts of the manuscript focusing on one or two main objectives.

The authors state that the desert dust event was unusal. In what sense? Why it was unusual? Please add some comments.

* P4 line 15: I doubt that the telescope field-of-view is 1.8mrad. Please verify and provide the correct value. 1.8mrad is the FoV of so-called x-ceilometers manufactured by Jenoptik, now Lufft.

* CL and CL* are not really constant. They change over time. There was a discussion within E-Profile and ToProf about the vocabulary. I suggest using calibration value instead of calibration constant.

* I dont understand why the authors focus on CL which is systematically biased by ignoring the two-way transmission of particles when their method includes the use of sun photometer measurements in order to estimate the correct lidar ratio. I would understand it in case that there are no co-located sun photometer measurements which is not the case for ICENET. The description of the method should be adapted. Either there are information about AOD and hence CL* could be derived or AOD is not available.

* p7 lines 3-11: it is missing a description about the treatment of lower 250 or 300m where the overlap correction is very large and often unreliable.

* the authors highlight that the best calibration values are obtained for low AOD. But then they derive calibration values during a desert dust episode with high AOD. This

is very strange for me. For me it would makes more sense deriving calibration values before and after the desert dust event at times with low AOD and interpolate them for the desert dust measurements. This would even enable comparisons for ceilometer and lidar during night-time.

* The ceilometers were measuring 5 days. The calibration was only performed on 11 half-hour intervals. It would be good if the method descrption includes a few sentences how the calibration values for each individual profile were derived. Example: Figure 7 shows that there is a calibration value for each profile.

* the unit of center of mass is missing. It should be m or km (e.g. p8 lines 23/24, but also on other parts of the manuscript Minor critical comments

* I found 5 times "Iberian Ceilometer Network (ICENET)". It is sufficient to explain the abbreviation one time

* P4 line 17: the authors wrote "the overlap is 90% complete at 555 m agl". The manufacturer provides an individual overlap function for each single ceilometer. I strongly doubt that all 5 ceilometers have the same overlap. I suggest providing a range.

* P4 line 35 "Raman-shifted channels" this formulation is a bit sloppy. The channels are not Raman-shifted. Re-frase!

* p5 line 11-12: please add the emitted laser energy P0 in the details of the calibration value.

* p5 line 23: the rcs signal is also normalized to the number of laser shots which vary from profile-to-profile

* p5 line 23/24: "and also, the overlap function of the instrument is factory determined." Although it is correct that the overlap function is determined by the manufacturer, I guess the authors wanted to point out that the overlap correction is already included in the rcs.

\* p5 line 28: "a region in the ceilometer profile". Here is missing the depth of the region. On page 7 the authors wrote about 990m. It should be mentioned already here.

\* p6 line 9: "low aerosol load, where the transmittance due to particles is close to 1". Please provide an AOD value what you consider as low. For instance the transmittance is 0.9 for an AOD of 0.1. Is this considered low? Or it is still too high?

\* p6 line 25 ff "continously calculating" Do you really mean continously or you mean repeatedly? And there are missing some details. I doubt that CL is calculated for every single profile. Do you use a temporal average for improving signal-to-noise ratio?

\* p7 line 3/4 "In this study 30-min profiles are used for comparison with lidar profiles." Is the calculation of the calibration value based on 30min too or a longer average is used?

\* p7 line 17: "has to be close to the CL" what is close? Maybe the author could write something like "has to be close to Cl within x%"

\* p11 line 21: "total attenuated backscatter" what is the difference between attenuated backscatter and total attenuated backscatter?

\* table 5: it is obvious when the site, the ceilometer and the sun photometer have the same geographical coordinates, that ceilometer and sun photometer are co-located. Hence it is not needed to write: co-located

\* Figure caption 1: Raman with capital R

\* Figure 2: middle image: unit of center of mass difference is missing. it's m or km?

\* Figure caption 3: Monte Carlo instead of Montecarlo

\* Figure 7: it's attenuated backscatter or total attenuated backscatter?

\* Figure 9: I suggest adding the sun photometer values in the top panel. As the lidar ratio was derived, it is straight forward to derive the integrated backscatter from sun

photometer AOD
* * *

---

## Author Comment (AC1) · 11 Aug 2017

we merged responses to referees in one single document. Please find in the supplement document a zip file with the responses and a new version of the manuscript

Please also note the supplement to this comment:
https://www.atmos-chem-phys-discuss.net/acp-2017-151/acp-2017-151-AC1-supplement.zip

---

## Author Response (AR1)

Dear editor and referees,

We would like to thank the referees for the valuable comments. There is a major concern that both referees share and we would like to address that in the first place, and then answer separately for each question.

This major concern refers to the calibration process, and not including the 2-way transmission (Tp). From this, it derives the concern of performing the calibration during the event if the aerosol load is high. We must say that the calibration process and obtaining backscatter coefficient profiles are two separate processes but thanks to the referees comments we realized that not including the Tp, the process is less intuitive and harder to explain. We recalculated everything including the Tp calculated with the AOD and values for the calibration factor for each site are provided. The entire methodology section has changed. Each process is described in a subsection (calibration on section 3.1 and inversion and validation on section 3.2 and 3.2.1). The distinction between CL* and CL is no longer necessary, so equations has been modified accordingly. We hope that this new structure provides a better understanding of the methodology applied.

We revised the manuscript according to the rest of suggestions and we replied to all the concern raised by the referees. Next paragraphs show comments and questions from referees (in red) and our response (in black).

Concerning Referee #1:

The authors state that the desert dust event was unusal. In what sense? Why it was unusual? Please add some comments.

Valenzuela et al., (2012) presented a classification of dust events affecting southern Spain during the period 2005-2010. They established 3 different typical patterns for the dust plumes corresponding with different synoptic scenarios. According to their results, only 22% of the desert dust events followed the same pattern described on Section 4.1.

The intensity of the event is also unusual for that time of the year. For this pattern, Valenzuela et al. (2012) reported a maximum AOD in Granada AERONET station of 0.98, being the pattern with the lower intensity of the three pattern (maximum AOD for the other pattern classifications were 1.6 and 1.4). During the dust event presented in this study, a maximum AOD of 1.77 was measured at Granada AERONET station (see new Fig. 6) which is significantly higher than the value reported by Valenzuela et al. (2012).

Also, there are two recent publications on the same event (Sorribas et al., 2017; Titos et al., 2017) that demonstrates the extraordinariness of the event. In particular, Titos et al. (2017) showed that the PM10 concentrations were very high, with 90% of stations exceeding the PM10 daily limit of 50 ug/m$^3$ (especially in southern Spain where the average PM10 was above 150 ug m$^3$. Also, at Montsec, the aerosol light extinction coefficient (from nephelometer + maap) at ground level was the highest measured in this station (Titos et al., 2017).

These references are now included in the reference list and on Section 4 we have included the following information:

*"Sorribas et al. (2017) studied the same event and compared it with meteorological parameters, aerosol properties and ozone from historical data sets on a site in southern Spain. They concluded that the event was exceptional because of its unusual intensity, its impact on surface measurements and the month of occurrence. In addition, Titos et al. (2017) also analyzed this event using 250 air quality monitoring stations over Spain to investigate the impact and temporal evolution of the event on surface PM10 levels. They also investigated aerosol optical properties, including attenuated backscatter from ceilometer during the event at Montsec station (one of the station included in ICENET). They concluded that the impact on surface PM10 was exceptional and highlighted the complexity of the event."*

And later on Section 4.1:

*"For dust events following a similar pattern that the one described here during the period 2005-2010, Valenzuela et al., (2012) reported a maximum AOD of 0.98 at Granada, which is significantly lower than the maximum measured at Granada station during this event."*

\* P4 line 15: I doubt that the telescope field-of-view is 1.8mrad. Please verify and provide the correct value. 1.8mrad is the FoV of so-called x-ceilometers manufactured by Jenoptik, now Lufft.

This was already changed on the previous revision of the manuscript:

*"The laser beam divergence is less than 0.3 mrad and the laser backscattered signal is collected on a telescope with a field of view of 0.45 mrad"*

\* CL and CL\* are not really constant. They change over time. There was a discussion within E-Profile and ToProf about the vocabulary. I suggest using calibration value instead of calibration constant.

Calibration value, factor or parameter is used instead of constant. In any case, CL\* is no longer used in the new version.

\* I dont understand why the authors focus on CL which is systematically biased by ignoring the two-way transmission of particles when their method includes the use of sun photometer measurements in order to estimate the correct lidar ratio. I would understand it in case that there are no co-located sun photometer measurements which is not the case for ICENET. The description of the method should be adapted. Either there are information about AOD and hence CL\* could be derived or AOD is not available.

As explained before, the entire methodology section has changed. The two-way transmission is now included.

\* p7 lines 3-11: it is missing a description about the treatment of lower 250 or 300m where the overlap correction is very large and often unreliable.

A constant value is considered:

*"The first 300 m of the profile are assigned to the value at 300 m to avoid very large overlap correction."*

\* the authors highlight that the best calibration values are obtained for low AOD. But then they derive calibration values during a desert dust episode with high AOD. This is very strange for me. For me it would makes more sense deriving calibration values before and after the desert dust event at times with low AOD and interpolate them for the desert dust measurements. This would even enable comparisons for ceilometer and lidar during night-time.

This is completely different with the new methodology section. Whenever the AOD matches the integral of a backscattering coefficient profile multiplied by the lidar ratio, a CL value is provided. The CL of a site would be an average value for a long-time series of calibrations. The aim is not to obtain calibrations during the event, but in order to obtain inverted backscattering coefficient profiles, at the reference height, the ration between RCS and molecular backscatter (Eq. 3) need to match.

\* The ceilometers were measuring 5 days. The calibration was only performed on 11 half-hour intervals. It would be good if the method descrption includes a few sentences how the calibration values for each individual profile were derived. Example: Figure 7 shows that there is a calibration value for each profile.

This has changed with the new methodology section. The calibration factor for each site is an average of a long time series of calibrations. The values for each site are provided on the new Table 2.

\* the unit of center of mass is missing. It should be m or km (e.g. p8 lines 23/24, but also on other parts of the manuscript Minor critical comments

The relative difference between the center of mass of each profile is provided, not the center of mass. The center of mass is shown on Fig. 9b (units are meters).

\* I found 5 times "Iberian Ceilometer Network (ICENET)". It is sufficient to explain the abbreviation one time

This has been corrected. Only abstract and introduction indicate both name and acronym.

\* P4 line 17: the authors wrote "the overlap is 90% complete at 555 m agl". The manufacturer provides an individual overlap function for each single ceilometer. I strongly doubt that all 5 ceilometers have the same overlap. I suggest providing a range.

A range is provided in the new manuscript:

*"the overlap is 90% complete between 555 and 885 m agl"*

\* P4 line 35 "Raman-shifted channels" this formulation is a bit sloppy. The channels are not Raman-shifted. Re-frase!

'Raman channel' is used instead.

\* p5 line 11-12: please add the emitted laser energy P0 in the details of the calibration value.

The emitted laser energy per pulse of the ceilometers is provided on Section 2:

"*The energy per pulse is 8.4 µJ with a repetition frequency in the range of 5 – 7 kHz*"

\* p5 line 23: the rcs signal is also normalized to the number of laser shots which vary from profile-to-profile

The manuscript has been modified:

"*the range corrected signal ($RCS(z) = P(z) \cdot z^2$), using an overlap function determined by the manufacturer and corrected for the number of laser shots*"

\* p5 line 23/24: "and also, the overlap function of the instrument is factory determined." Although it is correct that the overlap function is determined by the manufacturer, I guess the authors wanted to point out that the overlap correction is already included in the rcs.

That is correct:

"*the range corrected signal ($RCS(z) = P(z) \cdot z^2$), using an overlap function determined by the manufacturer and corrected for the number of laser shots*"

\* p5 line 28: "a region in the ceilometer profile". Here is missing the depth of the region. On page 7 the authors wrote about 990m. It should be mentioned already here.

We changed the manuscript accordingly:

"*we select regions of 990 m with a difference in gradients below 1%.*"

\* p6 line 9: "low aerosol load, where the transmittance due to particles is close to 1". Please provide an AOD value what you consider as low. For instance the transmittance is 0.9 for an AOD of 0.1. Is this considered low? Or it is still too high?

With the new structure of the methodology, we think this is no longer an issue.

\* p6 line 25 ff "continously calculating" Do you really mean continously or you mean repeatedly? And there are missing some details. I doubt that CL is calculated for every single profile. Do you use a temporal average for improving signal-to-noise ratio?

The section including those lines has changed completely. In any case, references to "continuously" has been changed. Also, with the new description of the calibration process it is clear that profiles are averaged for one hour.

\* p7 line 3/4 "In this study 30-min profiles are used for comparison with lidar profiles." Is the calculation of the calibration value based on 30min too or a longer average is used?

The calibration is performed on one hour averaged profiles. In order to compare with lidar, profiles for the inversion are averaged for 30 minutes.

* p7 line 17: "has to be close to the CL" what is close? Maybe the author could write something like "has to be close to Cl within x%"

This has changed with the new methodology. A range of mean±std of the calibration factor is used.

* p11 line 21: "total attenuated backscatter" what is the difference between attenuated backscatter and total attenuated backscatter?

They are the same, and manuscript has been changed to include 'total' in all cases. Total makes reference to particle + molecular.

* table 5: it is obvious when the site, the ceilometer and the sun photometer have the same geographical coordinates, that ceilometer and sun photometer are co-located. Hence it is not needed to write: co-located

Co-located has been removed from the table.

* Figure caption 1: Raman with capital R

Changed.

* Figure 2: middle image: unit of center of mass difference is missing. it's m or km?

It is a relative difference between the center of mass of both profiles. It has no units.

* Figure caption 3: Monte Carlo instead of Montecarlo

Changed.

* Figure 7: it's attenuated backscatter or total attenuated backscatter?

It is total attenuated backscatter. We changed the figure caption.

* Figure 9: I suggest adding the sun photometer values in the top panel. As the lidar ratio was derived, it is straight forward to derive the integrated backscatter from sun photometer AOD.

AOD and Ångström exponent are included in the new Figure 6 on section 4.1.

Concerning Referee #2:

1) The title does not clearly reflect the content of the paper. The input of sun photometer data is essential for the proposed method, but the title does not provide any hint that the described method is not usable for standalone ceilometer instruments. Further, the manuscript provides no proof, why the described dust outbreak event is extraordinary.

We modified the title to include a reference to sun-photometer data and we have removed the 'extraordinary' from the title:

"*Near real time processing of a ceilometer network assisted with sun-photometer data: monitoring a dust outbreak over the Iberian Peninsula*"

2) What makes the described event extraordinary? Is it the meteorological situation, intensity, duration or something else? If the event was extraordinary in a certain aspect, this statement should be proved by comparison with typical events. If the extraordinariness of the event cannot be corroborated, the title should be adopted.

Valenzuela et al., (2012) presented a classification of dust events affecting southern Spain during the period 2005-2010. They established 3 different typical patterns for the dust plumes corresponding with different synoptic scenarios. According to their results, only 22% of the desert dust events followed the same pattern described on Section 4.1.

The intensity of the event is also unusual for that time of the year. For this pattern, Valenzuela et al. (2012) reported a maximum AOD in Granada AERONET station of 0.98, being the pattern with the lower intensity of the three pattern (maximum AOD for the other pattern classifications were 1.6 and 1.4). During the dust event presented in this study, a maximum AOD of 1.77 was measured at Granada AERONET station (see new Fig. 6) which is significantly higher than the value reported by Valenzuela et al. (2012).

Also, there are two recent publications on the same event (Sorribas et al., 2017; Titos et al., 2017) that demonstrates the extraordinariness of the event. In particular, Titos et al. (2017) showed that the PM10 concentrations were very high, with 90% of stations exceeding the PM10 daily limit of 50 ug/m$^3$ (especially in southern Spain where the average PM10 was above 150 ug m$^3$. Also, at Montsec, the aerosol light extinction coefficient (from nephelometer + maap) at ground level was the highest measured in this station (Titos et al., 2017).

These references are now included in the reference list and on Section 4 we have included the following information:

"*Sorribas et al. (2017) studied the same event and compared it with meteorological parameters, aerosol properties and ozone from historical data sets on a site in southern Spain. They concluded that the event was exceptional because of its unusual intensity, its impact on surface measurements and the month of occurrence. In addition, Titos et al. (2017) also analyzed this event using 250 air quality monitoring stations over Spain to investigate the impact and temporal evolution of the event on surface PM10 levels. They also investigated aerosol optical properties, including attenuated backscatter from ceilometer during the event at Montsec station (one of the station included in ICENET). They concluded that the impact on surface PM10 was exceptional and highlighted the complexity of the event.*"

And later on Section 4.1:

"*For dust events following a similar pattern that the one described here during the period 2005-2010, Valenzuela et al., (2012) reported a maximum AOD of 0.98 at Granada,*

*which is significantly lower than the maximum measured at Granada station during this event."*

3) Why is the transmittance due to particles neglected? The authors describe how sun photometer data are used to constrain the Klett-Fernald inversions. Thus, all relevant information for the calculation of Tp is available. Why is it not used?

We agree with the reviewer. This has changed in the new methodology section.

4) The authors shall provide an estimation of the difference between CL* and CL for different particle optical depths.

With the new approach, there is no CL*.

5) It would be nice to have an example plot for illustrating the calibration method which is described at page 7. It would be even better to show two examples, one of a clear day and one of a polluted day.

With the new methodology section we hope that it is clear enough and that there is no need to include an additional plot.

6) The manuscript suffers from a major internal conflict. First, it is introduced that the calibration parameter CL can be retrieved only on days with low aerosol load. But in the next part, a period with very high aerosol load is used for demonstration and validation of the method. All CL values which are derived during the dust period will lead to a systematic bias of profiles if they are applied to measurements in clean conditions. The retrieved CL values are systematically too small due to neglecting Tp. To overcome this problem, the authors should derive and present a longer time series of CL values, with clean periods before and after the dust event. According to the theory, only the maximum values of CL in this time series (clean periods) are ok. Those maximum values before and after the dust event should be used for the retrieval of the attenuated backscatter profiles during the event. The lidar profiles can be used for the validation of these ('clean') CL values. But, it makes no sense to derive CL values during the dust event (even if constrained with lidar profiles) without taken into account Tp. All attempts to constrain CL values measured during the dust event should be removed from the manuscript. This includes Figures 2 and 3, and all text below equation 7 and the begin of section 4. Besides, the use of the correlation coefficient and center of mass as measures of goodness of the calibration seems to be of little help. Even if the calibration value is wrong, both profiles should have the same shape (and therefore the same Cmass and high R) because they are measured under the same atmospheric conditions.

The calibration and validation of inversion are independent processes and with the new structure of Section 3 (Methodology) we hope there is no doubt about the calibration process (not during the event) and the validation of the inversion during the event.
Figure 2 is one of the most important figures in the manuscript. Especially Figure 2a (modified to match the new methodology section) where it is shown that, at the reference height (Zref) which is a key parameter for the Klett-Fernald inversion, the ratio between the RCS and molecular backscatter (technically the same as CL in Eq. 3 in the manuscript) has to be similar to a long-term calibration factor calculated with the calibration method proposed. In particular, inversions that present at Zref this ratio between ± the standard

deviation of the mean calibration factor have smaller difference in NMB with the lidar profiles (considered the truth) so the long term calibration factor can be used to determine automatically if an inversion is good or not. The center of mass does not varies significantly with good or bad calibrations, which is good because highlights that the center of mass can be obtained directly from the total attenuated backscatter and no inversion is needed. However, the R does varies even the atmospheric conditions are the same (both instruments are measuring the same). The way the inversion algorithm distributes the backscatter coefficient with range depends on Zref, and if Zref is not correct, the inverted profiles differ from the lidar profile. On Fig. 2c better R values correspond with the values within the mean CL±std, and the worst case have a R of about 0.5 (a rejected profile).

7) The link between the two parts of the manuscript (methodology and results) is week.

With the methodology section this link between sections is stronger.

8) The description of the meteorological situation during the event is quite lengthy while some interesting measurement data are not provided, e.g. ,depolarization profiles from the lidar, time series of optical depth and Angstroem exponent, fine-to-coarse mode fraction etc. from the photometer network.

The intention if this manuscript is to describe the capabilities of the network of ceilometers for the monitoring of singular events such as the dust event described. Thus, the description of the event is aimed to show the path of the dust plumes reaching the Iberian Peninsula but we do not intend to give a full characterization of the event.

In any case, as indicated in previous questions we added some references and extra information about the event on Section 4:

"*Sorribas et al. (2017) studied the same event and compared it with meteorological parameters, aerosol properties and ozone from historical data sets on a site in southern Spain. They concluded that the event was exceptional because of its unusual intensity, its impact on surface measurements and the month of occurrence. In addition, Titos et al. (2017) also analyzed this event using 250 air quality monitoring stations over Spain to investigate the impact and temporal evolution of the event on surface PM10 levels. They also investigated aerosol optical properties, including attenuated backscatter from ceilometer during the event at Montsec station (one of the station included in ICENET). They concluded that the impact on surface PM10 was exceptional and highlighted the complexity of the event.*"

And later on Section 4.1:

"*For dust events following a similar pattern that the one described here during the period 2005-2010, Valenzuela et al., (2012) reported a maximum AOD of 0.98 at Granada, which is significantly lower than the maximum measured at Granada station during this event.*"

9) In general, the description of the event would be more useful if the authors could provide references to other studies about dust over the Iberian Peninsula. How often do events like the described dust outbreak occur at the Iberian Peninsula? What are typical

pathways and meteorological conditions? What are typical optical properties (e.g. optical depth) of the dust? What are typical geometrical properties of the dust layers? What makes this event special compared to others?

Valenzuela et al., (2012) presented a classification of dust events affecting southern Spain during the period 2005-2010. They established 3 different typical patterns for the dust plumes corresponding with different synoptic scenarios. According to their results, only 22% of the desert dust events followed the same pattern described on Section 4.1.

The intensity of the event is also unusual for that time of the year. For this pattern, Valenzuela et al. (2012) reported a maximum AOD in Granada AERONET station of 0.98, being the pattern with the lower intensity of the three pattern (maximum AOD for the other pattern classifications were 1.6 and 1.4). During the dust event presented in this study, a maximum AOD of 1.77 was measured at Granada AERONET station (see new Fig. 6) which is significantly higher than the value reported by Valenzuela et al. (2012).

On Section 4.1 we added the following information:

"*For dust events following a similar pattern that the one described here during the period 2005-2010, Valenzuela et al., (2012) reported a maximum AOD of 0.98 at Granada, which is significantly lower than the maximum measured at Granada station during this event.*"

10) abstract, the last sentence "... quantitative optical aerosol characterization with ceilometers..." is misleading. Ceilometers can be used for the quantification of the aerosol layering or aerosol load in terms of particle backscatter coefficients, but they cannot be used for characterization. 'Characterization' is often used as synonym for 'detection of aerosol type' or 'retrieval of intensive optical properties (like lidar ratio or Angstroem exponent)'. Ceilometers cannot provide this kind of information.

The term 'characterization' could be used on very different contexts depending on the instrumentation available, but we understand the referee concern. We used the term 'monitoring' in the title and the last sentence of the abstract was changed to:

"*Results reveal that it is possible to obtain quantitative optical aerosol properties (particle backscatter coefficient) and discriminate the quality of these retrievals with ceilometers over large areas*"

11) p2, l22: It is strange to write about the inadequate quality of satellite products in a paper about aerosol profiles from ceilometers. Certainly, products from ceilometers are very useful, but usually they do not provide profiles of higher quality than space-borne lidars.

We do not question the quality of satellite products and we never intended to suggest that ceilometers are better than satellite products. It is a fact that the temporal and spatial resolution of satellites is, sometimes, inadequate for specific applications. Also, validation of satellite retrievals often requires ground measurements. Those ground measurements do not need to be, and we do not intend them to be, coming from ceilometer. Thus, to avoid any doubt, we modified the sentence:

"*The main disadvantage of measurements from space-borne platforms is the low temporal resolution, since the measurements are limited to the satellite passes over a region*"

12) p2, l28: There are more relevant references, e.g. Flentje et al. 2010.

We included the reference.

13) introduction: The order of sentences in the last two paragraphs seems to be somewhat randomly. Maybe due to copy-and-paste? An outline to the structure of the manuscript is missing.

We modified the last part of the introduction and included an outline.

14) equation 1: CL* is not a constant in a strict sense. It changes on long time scales, e.g. due to laser aging or window contamination. Better to use 'parameter' instead of 'constant'.

All references to constant has been removed and 'value', 'factor' or 'parameter' is used instead.

15) p7, l29: how can negative CL values be explained? NMB is usually calculated as mean value of the bias profile ( b_ceil(z) - b_lidar(z) ) / b_lidar(z)

The reference height is calculated by Rayleigh fit automatically and, due to signal-to-noise ratio, sometimes the window selected to check the slope has, on average, a negative value. This is definitely a bad region for calibration, and therefore is rejected. The problem is when this reference height is automatically selected for the inversion algorithm. If this is the best Zref obtained, the inversion is not going to be valid. It is important to note, one of the findings here is that the calibration factor at Zref need to be around the value of the long-term calibration factor. This is what makes the automatic and unsupervised near-real time inversion possible.

16) Figure 5: This kind of trajectory plot seems to be less informative than the traditional plots (with a map projection and a time-altitude plot below). To which times of observation do the red dots refer to?

This figure is aimed to show the height intervals where African dust may have had an impact at two of the ceilometer network sites. The plots are in the form height vs latitude of back-trajectories, which is not the most common format but it is the simplest and clearer way to show what is intended. Please note that latitude and height are the relevant variables. The southern trajectories arrive from Africa (this is clear for Granada but also holds for Badajoz). The map projection would show the same information but it would be needed to use a color code for the changing heights along each trajectory, which makes the plots not so clear in the end. Time-latitude plots do not give the information we need to show.

The red dot in a trajectory corresponds to the air parcel's position at 12UTC. When this position is located to the north of the dust activation areas, then no dust is carried along with the air parcel even if that parcel came from northern Africa. This corresponds to the

height intervals depicted in green. Figure caption already indicated the meaning of the red dots (circles), but following the suggestion of the Reviewer we have explicitly added the corresponding time.

17) Figure 6: Is it really necessary to show this figure?

With the detailed description of section 4.2, we agree that the figure is not necessary. We removed it from the manuscript.

18) Figure 7: The labels of the color bar are unreadable. Time axes of the different stations have different tick scales.

We modified the Figure accordingly.

19) Figure 8: Where does the uncertainty of 15% comes from? Please, indicate the measurement times of the individual profiles by vertical lines in figure 7.

It comes from Fig. 2a. As we move away from the mean CL, the NMB increases and this value is used as an estimation of the uncertainty. We also added the vertical lines on Fig 7.

20) Figure 9: Please, add time series of AOD and columnar mean lidar-ratio (whenever Integrated bsc and AOD are available). Also time series of Angstroem exponents and fine-to-corase mode fractions would be interesting.

AOD and Ångström exponent is included in the new Figure 6 is section 4.1.

21) The readability of the text could be improved by splitting some long sentences into shorter ones, e.g.
* p1, l23+24
* p1, l25+26
* p3, l1+2
* p3, l18-20
* p5, l28-31

We looked at the manuscript carefully and modified some long sentences.

22) p1, l25: it is not clear to which method the term "this method" refers.

It is said at the beginning of the paragraph:

"*In this work we describe a method that uses aerosol optical depth (AOD) measurements from the AERONET network that it is applied for the calibration and automated quality assurance of inversion of ceilometer profiles*"

23) p1, l26: the date of the event is described several times in the manuscript as ... on 20 February and lasted until 24 February... -> it would be better to write "between... and .." or "... lasted from .. to..."

We modified that throughout the manuscript

24) p1, l21 + p2 l1: what refers "their" to?

The sentences has been modified:

"*The aerosol direct effects depend on the optical properties and spatial and vertical distribution of the aerosol in the atmosphere. In spite of the recent advances on instrumentation that has improved the ability of characterizing key aerosol properties and increase the spatial resolution, the associated uncertainties are still considered to be one of the majors in climate forcing (Boucher et al., 2013).*"

25) p2, l6: The terms "in-situ" and "surface measurements" are often used for ground based remote sensing instruments like lidars, in contrast to space-borne instruments. Maybe a term like "measurements of aerosol properties at ground level" could be used instead.

We changed the term to 'ground level aerosol measurements'.

26) p2, l7: when speaking about a covered area, it would be better to use "Europe" instead of "European Union".

Changed.

27) p2, l10: (and elsewhere in the paper): vertical resolved -> vertically resolved

Changed

28) p2, l31: make complicate -> hinder?

Changed

29) p3, l18-20: the calibration is used to validate the inversion?

Changed

30) p4, l2: comparable -> calibrated?

Changed

31) p4, l4: constraint -> constrain

Changed

32) p5, l16: not overlap -> no overlap

Changed

33) p5, l28: what means "computing the Rayleigh fit"? -> ... particle free regions are determined by comparing the gradient... If the difference is below 1%, we can assume particle free conditions...

We changed the sentence:

"*The Rayleigh fit, compares the gradient with altitude (the slope) of both profiles, and looks for a region in the ceilometer profile that has the same trend than the expected molecular profile*"

34) p7, l12: ... has the influence of ... -> ...is influenced by...

Changed

35) p10, l26 ... northern African... -> ... northern Africa...

Changed

**References:**

[revised manuscript text omitted]